# Microfluidic Chip with Low Constant-Current Stimulation (LCCS) Platform: Human Nucleus Pulposus Degeneration In Vitro Model for Symptomatic Intervertebral Disc

**DOI:** 10.3390/mi12111291

**Published:** 2021-10-21

**Authors:** An-Gi Kim, Tae-Won Kim, Woo-Keun Kwon, Kwang-Ho Lee, Sehoon Jeong, Min-Ho Hwang, Hyuk Choi

**Affiliations:** 1Department of Medical Sciences, Graduate School of Medicine, Korea University, Seoul 08308, Korea; 7.437.280@gmail.com (A.-G.K.); verybest1007@korea.ac.kr (T.-W.K.); 2Department of Neurosurgery, College of Medicine, Korea University Guro Hospital, Seoul 08308, Korea; kwontym@gmail.com; 3Division of Mechanical and Biomedical, Mechatronics, and Materials Science and Engineering, College of Engineering, Kangwon National University, Chuncheon 24341, Korea; khmhlee@kangwon.ac.kr; 4Department of Healthcare Information Technology, Inje University, Gimhae 50834, Korea; jeongsh@inje.ac.kr

**Keywords:** microfluidic chip, electrical stimulation, low constant-current stimulation, intervertebral disc degeneration, inflammation

## Abstract

Intervertebral disc (IVD) degeneration is a major cause of low back pain (LBP) in the lumbar spine. This phenomenon is caused by several processes, including matrix degradation in IVD tissues, which is mediated by matrix metalloproteinases (MMPs) and inflammatory responses, which can be mediated by interactions among immune cells, such as macrophages and IVD cells. In particular, interleukin (IL)-1 beta (β), which is a master regulator secreted by macrophages, mediates the inflammatory response in nucleus pulposus cells (NP) and plays a significant role in the development or progression of diseases. In this study, we developed a custom electrical stimulation (ES) platform that can apply low-constant-current stimulation (LCCS) signals to microfluidic chips. Using this platform, we examined the effects of LCCS on IL-1β-mediated inflammatory NP cells, administered at various currents (5, 10, 20, 50, and 100 μA at 200 Hz). Our results showed that the inflammatory response, induced by IL-1β in human NP cells, was successfully established. Furthermore, 5, 10, 20, and 100 μA LCCS positively modulated inflamed human NP cells’ morphological phenotype and kinetic properties. LCCS could affect the treatment of degenerative diseases, revealing the applicability of the LCCS platform for basic research of electroceuticals.

## 1. Introduction

Intervertebral disc (IVD) degeneration is the primary factor causing low back pain (LBP) [1,2] and has been reported to occur due to factors such as senescence, mechanical stress, and genetic factors [3,4,5,6]. Over 600 million people worldwide suffer from LBP, and currently, an astronomical amount of medical expenses are being incurred by various countries to treat patients with IVD disease, thereby increasing the burden on socioeconomic expenses [7,8].

Histologically, IVD consists of external annulus fibrosis tissue and internal gelatinous nucleus pulposus cells (NP). Additionally, the IVD is composed of extracellular matrix (ECM) with collagen and fibrocatilaginous tissue. The ECM remodeling is believed to be regulated by matrix metalloproteinases (MMPs) and tissue inhibitors of metalloproteinases (TIMPs). The vascular and free nerve endings are extended to the 1/3 point of the external fibrosus in normal IVD structures, and other tissues are in an avascular and insensitive state. However, new blood vessels and nerve growths toward the internal IVD have been clinically observed in patients with lumbar pain [6,9,10]. In addition, many macrophages are observed in IVD tissue and are accompanied by abnormal inflammatory responses. In particular, interleukin (IL)-1 beta (β), released from invasive macrophages, acts as a master regulator of this inflammatory response and might play an important role in the pathological mechanism of IVD degeneration [11,12,13,14]. Together, the vascularization and nerve ingrowth are believed to be regulated by the ECM matrix breakdown and abnormal expression of inflammatory mediators, resulting from IL-1β-mediated inflammatory response. Additionally, these molecules result in the formation of a physical space and/or physiological response, which allows the ECs to invade into the centrally located NP region. However, no evident pathological and disease progression mechanisms for lumbar pain have been proposed. Basic research on IVD degeneration and its mechanism is not available due to limitations in advancing research levels through human in vitro models, including disease similarities and research reproducibilities.

Lab-on-a-chip technology, based on microfluidic engineering, has been developed to provide quantitative results similar to the physiological activity of the body in an in vitro model and is potentially applicable for the development of a new platform for various disease studies [15,16,17]. Cell co-culture platforms, based on microfluidics, can be used to analyze multicellular interactions in real-time and can be applied as an experimental tool that simulates the 3D microenvironments in the body. These platforms are also effective in the study of cell-to-cell communication because they simulate cellular responses and paracrine signaling by potential contributing factors, released from multiple or specific cells, that contribute to the generation and progression of diseases. Recent studies have been conducted on single cells under 2D culture environments [18,19,20]. However, the actual body forms various tissues, including 3D multicellular and extracellular matrices. Therefore, pathological mechanisms and therapeutic research for diseases should be performed in a 3D micro-environment with multicellular interaction [17,21,22,23,24].

In addition to the above requirement, some experimental approaches are available for treating various diseases in terms of engineering elements [18,25]. “Electroceutical” is defined as a technology or tool used to treat diseases by exerting an electrical signal or stimulation on the body. Similar to vagus nerve stimulation, it stimulates nerve circuits of the body that are connected to the brain to treat various diseases, such as arthritis, epilepsy, depression, and Crohn’s disease [26,27,28,29,30]. In addition, many studies have attempted direct electrical stimulation (ES) of in vitro environmental cells to determine a method for treating diseases, providing relief from inflammation and pain, or stimulating tissue regeneration [18,19,20]. Treatment with existing drugs can lead to adverse effects in unwanted regions via blood circulation of corresponding formulations, but an electroceutical can selectively stimulate specific tissues or nerves necessary for treatment. Because of the low cost of developing tools and the potential to control the expression of various genes, and their functions within cells, through ES, electroceuticals can be applied to the treatment of various diseases [18].

Accordingly, we developed an integrated microfluidic platform in which multiple types of cells could be incubated in 3D through the collagen hydrogel. It can simulate chemical and biological phenomena among cells such as autocrine, signaling across gap junctions, and paracrine effects via intercellular diffusion. It can also combine a tool to permit electrical signals. In this platform, we conducted the IL-1β mediated IVD degeneration in vitro model through diffusion paracrine. In addition, we examined the effect of ES on this IVD degeneration model.

## 2. Materials and Methods

### 2.1. Design and Structure of Low Constant-Current Stimulation (LCCS) Platform

#### 2.1.1. Microfluidic Chip Fabrication and Experimental Setup

We designed a microfluidic platform, comprising three distinct structures, connected through a micro-patterned channel array using standard soft lithographic techniques. The three chambers of the microfluidic platform had an incubation chamber that could incubate different cells at both ends, and a collagen-hydrogel channel connected to two chambers. In particular, multiple posts were placed in the middle collagen-hydrogel channel to create a 3D gel block for gel injections (Figure 1). This prevented leakage of collagen-hydrogel solution into the incubation chamber and created a gel block in the designated places. Each chamber and channel had holes for injecting various reagents, including an incubation batch.

To improve the cohesiveness with collagen hydrogel, before incubation, and organize the condition to allow easy attachment of cells within the chip, incubation at 37 °C for a minimum of 4 h was conducted after filling 1 mg/mL of poly D-lysine solution (PDL, Sigma-Aldrich, St. Louis, MO, USA). Then, the PDL solution was absorbed, and its residues were removed by washing with distilled water. Gelation was completed by 30 min incubation at 37 °C after injecting collagen hydrogel into the middle channel. Human NP cells were injected in a cell incubation chamber, incubated, and stabilized in an incubator for 30 min.

#### 2.1.2. Fabrication of the LCCS Platform

The LCCS platform consists of three duplex types and was manufactured using polylactic acid (PLA) through customized production. The first layer plays a role as an anchor of the whole system. The rest of the layers horizontally dock in the first layer, making the LCSS platform hold securely and level it off. The second layer provides a space for connecting nine multiple microfluidic chips (3 × 3). In the third layer, platinum electrodes are installed to circulate electricity, and the height of each electrode is designed to permit ES on the medium filled through the inlet/outlet of the incubation channel in the microfluidic chip (Figure 1a,b). After inserting the electrode in the hole in the third layer, nickel magnetic pins connected to supply signals from the LCCS are maintained in a combination of platinum electrodes and magnetic adhesion. The electrical connection is designed such that the signal, through the electrode and pin, with this connection is not interrupted. Additionally, the installation of a holding cap on the magnetic pin can prevent the separation of electrodes during movement or delivery (Figure 1c). The humidity of the internal platform, containing incubated microfluidic chips, is maintained by applying a dose of 5–10 mL sterilized ultra-pure water into the space between the first and second layers (Figure 1d). The type 1 collagen solution was perfectly filled into the hydrogel channel and stopped by surface tension between the posts. The hydrostatic pressure difference between the cell culture channel and scaffold channel enables interstitial flow. The cells consequently attach to the exposed surface of the collagen gel (Figure 1e,f).

#### 2.1.3. Constant-Current Generating Circuit of the LCCS controller

The LCCS device and controller (Figure 2a,b) consist of an overall LCCS platform connected by a wire connector. Each wire connector is a type of bundle six-wire connector that connects six wires supplying alternating current (AC) signals to three microfluidic chips (Figure 2c).

A function generator (10 MHz Sweep/Function Generator FG8210; DAGATRONICS CORPORATION, KOR) was applied, as a tool, to supply the LCCS platform with raw signals, and an Arduino board, installed with a microcontroller unit (MCU), was installed in the circuit. Since we require a constant current regardless of load impedance changes for biological testing, we are applying a low constant current for this experiment. The internal circuit of the LCCS controller has a repetitive process consisting of two different modes and was designed to maintain a constant current using a digital potentiometer (MCP4151-104E/P; Microchip Technology, Chandler, AZ, USA). The LCCS controller performs a repetitive process of switching to Measure & Control mode (M&C mode) every 0.1 s (10 Hz) while the Alternating mode is maintained. In the Alternating mode, biphasic AC supplied from the function generator stimulates medium and cells in the chips, and the M&C mode causes the low direct current (DC) to flow into the digital potentiometer for 0.003 s, where its impedance changes are measured and controlled. The LCCS controller has five resistors connected to one chip, along with two loads of the microfluidic chip and the digital potentiometer. In addition, general-purpose input/output (GPIO) ports that play an essential role in controlling the two modes are connected between the resistors, respectively. Only one resistor, connected in front of the potentiometer, was 500 Ω, and the remaining resistors were 5 kΩ. All resistors in the circuit were designed to reduce all supply signals during M&C mode. In particular, the 5 kΩ resistor plays a role in blocking the function generator’s supply signal, and the 500 Ω resistor serves to weaken the 5 V signal supplied from the GPIO Output port. In Alternating mode, all GPIO ports are changed to the Input state so that each pin has a high resistance, which is the same as the closed-circuit design in which the voltage divider does not operate. Therefore, the remaining signals, except for the voltage to take 5 kΩ resistors among the biphasic AC signals supplied by the Function generator, flow generally to the cell-cultured microfluidic chip. Conversely, in M&C mode, the GPIO port, connected in series with the 500 Ω resistors, is changed from input to output high to supply a DC signal of 5 V, and the remaining GPIO ports are changed from input to output low to form a single circuit. The supplied 5 V signal is lowered to ≈500 mV by 500 Ω resistors, and GPIO output low ports the voltage divider. The impedance of these low DC signals flowing through the potentiometer is measured by the analogue-to-digital converter (ADC), (ADS1115; TEXAS INSTRUMENTS, Dallas, TX, USA). In this case, since the AC signal supplied from the function generator undergoes two steps: 5 kΩ resistor, and a voltage divider composed of the output low (≈50 Ω) pins, its signal is weakened by about 1/1000 times. Therefore, it is designed not to significantly impact ADC measurements because it becomes weaker than the low DC signal. Therefore, it is designed not to significantly impact ADC measurements because it becomes weaker than the low DC signal. A digital multimeter (FLUKE 87-5; Fluke Corporation, Everett, WA, USA) was additionally installed to verify the AC of the medium in real-time. Hence, the operator can place an order or check the signal changes in real-time through the terminal window of the PC user interface (Figure 2d,e).

#### 2.1.4. Simulation Settings

A simulation with physical simulation software (COMSOL Multiphysics 5.6; COMSOL INC., Stockholm, Sweden) was performed to analyze the internal electrical signals of the incubation channel, which was supplied through the electrode. The circumstance of cell incubations was identified at levels of kΩ with resistance measurements and voltage ADC analysis using a digital multimeter (FLUKE 87-5; Fluke Corporation, Everett, WA, USA); therefore, the microfluidic impedance within the simulation channel was set to the same value. Similar to the actual experiment, four electrodes were arranged by inserting them into the inlet and outlet of the two incubation channels, based on one chip. The electrical potential and electric field were analyzed under the condition of ≈20 μA (≈300 mV) biphasic signal as an assumed impedance of the culture medium, based on the upper value (≈15 kΩ).

#### 2.1.5. Voltage ADC Analysis and Operation Assessment of the LCCS Controller

The impedance of the microfluidic chip in an internal channel, which is changed by permitting electrical signals, was analyzed. The ADC value—which transformed the existing circuit into two modes—was applied to verify the normal operation of the constant-current stimulator. The first process was to set up the condition that an internal chip cannot accept biphasic signals by stopping the operation of a function generator and the power supply. We defined ‘None’ as the status that no biphasic signals flow in the whole system. The value of the digital potentiometer was fixed as ≈50 kΩ since we do not need the LCCS to function. The M&C mode of the LCCS controller, however, keep on estimating the ADC value, repeatedly. Accordingly, the low monophasic signals can circulate in the medium every 0.1 s (10 Hz) with a value of ≈3% in the duty cycle. Conversely, the second process involves operating the LCCS function to again reconnect the functions of the function generator and digital potentiometer. At 200 Hz, 20 μA of biphasic signal flows within the internal chip. The condition of a low monophasic signal passing every 0.1 s with the M&C mode is defined as LCCS (Figure 2e,f). Based on these two conditions, the experiments were divided into naïve NP and inflamed NP groups, representing non-inflammatory and inflammatory naïve NP cells, respectively.

Furthermore, ADC analysis included a process in which an analog voltage, measured from the lateral stage of a digital potentiometer, is transferred to a digital value of 200 Hz at 20 μA on the circuit.

### 2.2. Biological Analysis and Platform Validation Setup

#### 2.2.1. Human NP Cell Culture

Human NP cells were isolated from the disc tissues of seven patients with degenerative spinal disease (Pfirrmann degenerative grades II–III) during elective surgery. This study was approved by the Korea University Hospital Institutional Review Board (KUGH170208-001), and informed consent was obtained from the subjects. All methods were performed in accordance with the guidelines and regulation of the Human Ethics Committee of the Korea University Hospital. Human NP cells were cultured in a nutrient mixture F-12 (Gibco-BRL, Grand island, NY, USA) supplemented with 10% fetal bovine serum (FBS; Gibco-BRL, Grand island, NY, USA) and 1% penicillin/streptomycin (P/S; Gibco-BRL, Grand island, NY, USA). After two days, the human NP cells were plated onto 75 cm^2^ culture flasks containing F-12, supplemented with 1% FBS and 1% P/S, at a density of 5 × 10^5^ cells per flask. Human NP cells were used at passage two.

#### 2.2.2. IL-1β Stimulation on Human NP Cells in The Microfluidic Chip

To mimic the inflammatory process of disc tissues in the early stage of intervertebral disc degeneration, IL-1β, which is secreted by immune cells, was applied as a master pro-inflammatory factor. The cultured medium was treated with 1 ng/mL IL-1β on the opposite channel of where human NP cells were incubated. The stimulation time was determined by the point at which more than 90% of the incubated cells turned from chondro-like cells to spindle-like cells. To inspect the changes in mRNA expression, cells were stored at −80 °C after collection and preparation. Various ES parameters (5, 10, 20, 50, and 100 μA) were applied to cells for 6 h, and the cells were cultured for an additional 18 h without ES.

#### 2.2.3. Immunocytochemistry (ICC)

Human NP cells in the microfluidic device were fixed with 4% paraformaldehyde and then permeabilized with 0.2% Triton X-100 in phosphate-buffered saline for 10 min, blocked with 3% bovine serum albumin for 1 h at 25 °C (room temperature), and then incubated with Alexa 488-conjugated phalloidin (Invitrogen, Carlsbad, CA, USA). Finally, the cells were counterstained with 4′,6-diamidino-2- phenylindole (Santa Cruz Biotechnology, Dallas, TX, USA). The samples were imaged using an EVOS FL auto cell imaging system (Thermo Fisher Scientific Inc., Waltham, MA, USA). In addition, for kinetic analysis, the computational imaging process included a target cell set up by a computational edging image, based on raw IF images obtained from the experiment, and various kinetic data were derived from the edge border setting after automated rendering. All imaging analyses were performed using the Icy imaging processing software (version 2.2.0.0) and ZEISS LSM 9 Zen-blue edition imaging software (version 3.2, CarlZeiss Microscopy GmbH, Niedersachsen, Germany).

#### 2.2.4. Semi-Quantitative Real-Time Polymerase Chain Reaction (sqRT-PCR)

Human NP cells were lysed with TRIzol reagent (Invitrogen, Carlsbad, CA, USA). The RNA was extracted, and cDNA was synthesized according to the manufacturer’s instructions (Life Technologies, Carlsbad, CA, USA). sqRT-PCR was performed to determine the mRNA levels of IL-8, MMP-1, and MMP-3 using the SYBR Green PCR Master Mix (Applied Biosystems, Waltham, MA, USA). mRNA expression was analyzed using the 2^−∆∆Ct^ method.

#### 2.2.5. Cell Cytotoxicity and Lactate Dehydrogenase Assay (LDH)

Lactate dehydrogenase (LDH) release was measured according to the manufacturer’s instructions (Roche, Basel, Switzerland). After the cells were exposed to IL-1β with/without ES, the exposure medium was collected to quantitate the release of LDH. In addition, naïve NP cells were examined as a positive control. Viability was calculated concerning that of the control (human NP cells treated with IL-1β). If human NP cells were damaged by ES, these cells would show a tendency toward increased LDH production.

#### 2.2.6. Statistical Analyses

Data are expressed as the mean ± SEM for three technical replications and four individual experiments using independent cell cultures. One-way analysis of variance and Bonferroni’s correction post hoc test were used to assess the differences among the experimental groups. All statistical analyses were performed using the SPSS software (version 21.3, IBM SPSS Statistics Inc., Chicago, IL, USA). Statistical significance was set at *p* < 0.05.

## 3. Results

### 3.1. Simulation of Electric Potential and Electric Field in Microfluidic Chip

If four electrodes with polarity (+, −) are inserted into each chamber, and the signal at 300 mV and 200 Hz is permitted, the simulation result indicates that the electric field magnitude is different for narrow or wide types of channels. In the case of Points 1 and 3 through a single channel, a high electric field is created, but the electric field at Point 2, which is faced with a scaffold channel, is slightly decreased by enlarging the channel. Because identical signals are permitted into two incubation channels, the same electric field was created at Points 2 and 5. However, the electric field could be created at Point 4 through a space dedicated for fluid exchange between posts because the point with the post had no electric fields, so the current could not exist. Therefore, the result of this simulation indicates that the strength of the electric field I Point 4 is slightly lower than that of the passage section (Points 2 and 5), where two electrodes are directly connected. In addition, cells that were incubated in a culture channel and transferred to the scaffold channel could be influenced by current (Figure 3a).

Figure 3c shows the electric field intensity and current direction created by potential differences between the two electrodes. Because the AC signal is permitted by the polarity changes of an electrode in the incubation chamber, the direction is transitioned over different polarities, implying a changed current direction (Figure 3b,c).

### 3.2. ADC Measurements and Constant-Current Operation Assessment

Based on the point, emphasized in this study, that LCCS is significantly influenced by deviations of load impedance connected to the circuit, we analyzed the relative impedance change for all four groups (naïve NP, naïve NP with LCCS, inflamed NP, and inflamed NP with LCCS). The corresponding values indicate the change rate of NP cells for each group from initial (0 min) impedance to 6 h (360 min) passed with or without LCCS. The standard value, 1.0, indicates no changes. In conclusion, the impedance after 6 h was significantly decreased in both naïve NP and inflamed NP with LCCS, and its decrease in naïve NP was larger than that in inflamed NP (Figure 3d).

To verify the impedance change and LCCS operations according to time, changes in naïve NP with LCCS and inflamed NP with LCCS were demonstrated every 30 min. The corresponding values represent the change rate of impedance measured at each point based on the initial impedance (at 0 min). As mentioned above, the impedance decrease in naïve NP was larger than that in inflamed NP as a function of time. Both groups showed rapid decreases in impedance at 1 h (60 min) and were observed every 10 min from 0 min to 1 h (60 min). Thus, the impedance change in naïve NP was stable after a rapid decrease within 40 min, but inflamed NP was stable after a rapid decrease within 30 min.

In addition, the symmetrical change, based on a standard value (1.0), indicates that the LCCS is normally operated. To maintain a constant current in a single circuit, the impedance of the digital potentiometer must also be changed by every constant decrease in NP cells (Figure 3d). Therefore, as the impedance change in the NP cells was constantly decreased to maintain the total impedance of the circuit, a symmetrical graph showed a constant increase in the digital potentiometer (Figure 3e).

### 3.3. Diffusion of Pro-Inflammatory Cytokine IL-1β and Mimicking the Degenerative Condition on Human NP Cells

To simulate an inflammatory model by immune cells in IVD degeneration, morphological changes, and mRNA expression changes of IL-8, MMP-1, and MMP-3 as inflammatory mediators were confirmed by immunofluorescence and qRT-PCR after treating IL-1β in the nucleus pulposus with diffusion paracrine.

Human NP cells were inserted into the cell culture chamber and treated with 1 ng/mL of recombinant IL-1β in the opposite channel connected to the collagen channel. The prepared IL-1β diffused into the opposite channel, which contained incubated NP cells over time, and caused an inflammatory response. From the immunofluorescence (IF) image, we observed that the cytoplasm of NP cells gradually shrank in morphology after 16 h of diffusion and that lamellipodia and filopodia were elongated and transitioned from chondrocyte-like cells to spindle-like cells. Over 48-h stimulation, the cytoplasm of human NP cells completely shrank with filopodia extension compared to other groups (Figure 4a). A significant increase in expression was statistically indicated in IL-8, MMP-1, and MMP-3 after 48 h diffusion, compared to naïve NP cells (Figure 4b).

These results indicate that the inflammatory response of human NP cells can be induced by diffusing IL-1β in our microfluidic platform.

### 3.4. Effects of ESon Inflamed Human NP Cells in the LCCS Platform

As a therapeutic approach for intervertebral disc degeneration, ES in various ranges (5, 10, 20, 50, and 100 μA) was conducted to inspect its effect through LCCS that was developed for human NP cells induced by IL-1β inflammation. The morphological change was observed by IF imaging, and kinetic analysis was completed through a computational imaging process (Figure 5a).

The IL-1β treatment group showed cytoplasmic shrinkage and filopodia extension in the IF image. The cytoplasm area in all permit parameters, except for the 50 μA group, recovered to the level of naïve NP cells in the stimulation-permitted groups (Figure 5b). Moreover, compared to the naïve NP cells, the results of the kinetic analysis showed statistically significant decreases in area, perimeter, and maximum Feret diameter for the IL-1β group and the other groups, except for the 50 μA group, indicating that each parameter was recovered to naïve levels. Nevertheless, only the 50 μA group showed a decrease in convexity, whereas the remaining groups did not exhibit a statistically significant difference in sphericity, elongation, and roundness (Figure 5c).

The overall results of IF imaging and kinetic analyses show that human NP cells undergo morphological changes to the type of spindle-like cells, under inflammatory circumstance by IL-1β simulation, and exhibit filopodia extension. Therefore, ES represents morphological regenerative effects in human NP cells under IL-1β-mediated inflammatory circumstances.

### 3.5. Cells Viability/Cytotoxicity (Live and Dead Assay)

Live/Dead viability/cytotoxicity assay is a common method used in cytotoxicity assays. Because ES can damage cells, we tested ES at a dose of 100 μA, which is the maximum dose used in this study. As shown in Figure 6, Human NP cells with ES, applied at 100 μA, and IL-1β treatment did not show a difference in cell viability/cytotoxicity compared to naïve NP cells (Figure 6).

## 4. Discussion

An important element of the pathomechanism for intervertebral disc degeneration is the interaction between two types of cells, including the external fibrosus, nucleus pulposus, and immune cells as non-intervertebral discs. In particular, the interaction between macrophages, owing to the nucleus pulposus and blood vessels, is considered the primary cause of back pain after initial degeneration. Many macrophages and inflammatory cytokine increases were observed in patients suffering from degeneration accompanied by back pain. In addition, a recent study focused on interactions with non-intervertebral cells, including blood vessels and nerve cells, as an important factor in the mechanism of degeneration and back pain. Even with various preliminary studies, the degeneration mechanism of IVD, accompanied by back pain, has not been clearly revealed. Therefore, alternatives for surgical treatments for these diseases, such as drugs or other types of treatment, have not been developed. The critical reason of this phenomenon is the absence of a platform to establish a disease model, including disease similarity, such as intercellular interaction, paracrine, and chemotactic effects; high experimental convenience, such as research reproducibility. In addition, no platforms in advanced models are available to expand the treatment.

To overcome these limitations in this study, we developed a microfluidic chip that can co-culture multiple cells and provide high-throughput for paracrine effects through the release of chemokines and pro-inflammatory cytokines. Furthermore, a combined platform was developed to determine the effect of ES as an approach for treatment. This platform can provide a more accurate and reliable in vitro model for various disease mechanisms related to multiple cells.

The two important elements for stimulation permission in this study are to set up a reference power source (constant-voltage/constant-current) and waveform (monophasic/biphasic). The two important elements for this study’s ES are setting up a reference power source (constant-voltage/constant-current) and waveform (monophasic/biphasic). The constant voltage signal induces variations in the current values on the channel by varying the channel impedance. However, the constant current signal provides a constant current to the channel without any impedance differences. Various research groups can collectively establish a standard reference profile on identical target cells, excluding different incubation conditions. This study was conducted using a constant current signal, for signal supply, to enhance the accuracy and uniformity of the experimental data.

Even with many preliminary studies on the ES effect of monophasic waveform [31,32], monophasic signals cause irregular current flow, due to accumulated proteins on the negatively charged surface, or decrease cell viability by increasing the pH of the batch, due to oxidation [33,34]. In contrast, biphasic signals can decrease protein accumulation and oxidation on the electrode surface, because of the rapidly alternating electrode polarity within channels, through high frequency [35,36,37]. Therefore, biphasic signals were applied to the cells in this study.

Circulating immune cells enter the damaged IVD during its degeneration and release various pro-inflammatory cytokines, including TNF-α and IL-1β. In particular, an important role of IL-1β is that it is a master regulator of IVD degeneration. This cytokine has several common functions, including chemoattraction of neutrophils, induction of adhesion molecule expression on endothelial cells, and stimulation of phagocytosis by macrophages [35,36]. Increased expression of IL-1β was clinically observed in degenerated and herniated intervertebral discs. Additionally, its expression promotes the expression of various ECM-modifying catabolic enzymes, including ADAMTS-4, -5, MMP-1, MMP-3, and MMP-13. MMP-1 and MMP-3 interact with collagen type 1 and type 2 or proteoglycans in the body, leading to a breakdown of ECM components. As a chemoattractant related to immune cell recruitment, IL-1β increases the expression of CCLs, chemokine ligand (CXCL), and IL-8 [37,38,39,40]. Our previous study also confirmed the increased expression of MMP and IL-8 [20,41,42,43,44,45]. When stimulation by IL-1β diffusion, within the LCCS platform, was permitted to incubate channels, this study also verified the increased gene expression of IL-8, MMP-1, and MMP-3 in human NP cells. The pathogenesis of IVD degeneration encompasses modification of the ECM and morphological and/or kinetic changes in IVD cells. These morphological and kinetic changes can be induced by damage to matrix-cell interactions in IVD tissue, and ECM homeostasis can be mediated by ECM-modifying enzymes, as described earlier. Therefore, morphological and kinetic changes in human NP cells, in this study, might be varied by MMP catabolic enzymes released from IL-1β inflammation and by inflammatory mediators, including ILs.

We induced inflammation and degeneration by IL-1β in the LCCS platform and evaluated the treatment effects of ES. ES, in the range of 5, 10, 20, 50, and 100 μA at 200 Hz, were permitted for human NP cells. The biological activation range of the microcurrent in the body was controlled to less than 300 Hz. In addition, we performed this study with a fixed frequency of 200 Hz because of optimal effects at this frequency. The ES for inflammatory factors was evaluated in the range of 100–300 Hz, as in a previous study [43]. The results of IF and kinetic analysis for ES, in human NP cells, confirmed the positive effects in the 5, 10, 20, and 100 μA groups treated with IL-1β and the recovery of kinetic properties at the level of naïve NP cells. However, the 50 μA group depicted no effect. With regard to this, differential dose responses can be represented by target proteins or genes rather than the permitted dose for its effects directed by their certain value. Because individual dose thresholds belong to specific proteins and genes, this phenomenon results from the concept that the corresponding factors are not activated without permitting these respondent doses. Furthermore, an excessive dose for a specific factor can indicate inhibitory effects or negate the beneficial response.

Modulation mechanisms in various tissues, including intervertebral disc by ES, are not clearly identified. However, in other predictable hypotheses, through previous research bases to which ES was applied, an important factor in the morphological change in NP cells is thought to be the regulation of ECM-modifying engines and inflammatory mediators [46]. Regulation of these factors can generally be induced through changes in MAPK and NF-kB signaling pathways. There is a previous study in which ES can control the expression of each factor by controlling MAPK and NF-kB [41,46,47,48]. Based on this, it can be effective in regeneration by lowering the inflammatory expression of NP cells in the IVD degeneration process, ultimately reducing the irritation of intervertebral nerves and lowering the expression of pain-related receptors and factors.

Therefore, it is more desirable to collect electrical profiles for proteins or genes with target cells during ES. The ES parameters used in this study may not be applicable in clinical practice. Because ES must be delivered to the target tissues or cells with sufficient energy, exploring the optimal dose is likely required for clinical application.

Nevertheless, to simulate the body through multiple cell co-culture, or 3D culture with human samples, and to permit various parameters of ES in high-throughput, system developments must be applied to various fundamental studies for ES.

## 5. Conclusions

In summary, we developed an integrated microfluidic platform for 3D co-culture to facilitate ES. We tempered a digital potentiometer for a secure constant current using MCU and IC. We also evaluated the changes in impedance and stability of the supply signal through an analytical process. Consequently, we validated that the constant current signal circulated in the whole system. In addition, we examined the effects of LCCS on IL-1β-mediated inflammatory NP cells. Hence, the platform developed in this study can establish an advanced model to determine the pathological mechanism of various diseases and provide easy circumstances for high-throughput cell response research with various ranges of ES.

## Figures and Tables

**Figure 1 micromachines-12-01291-f001:**
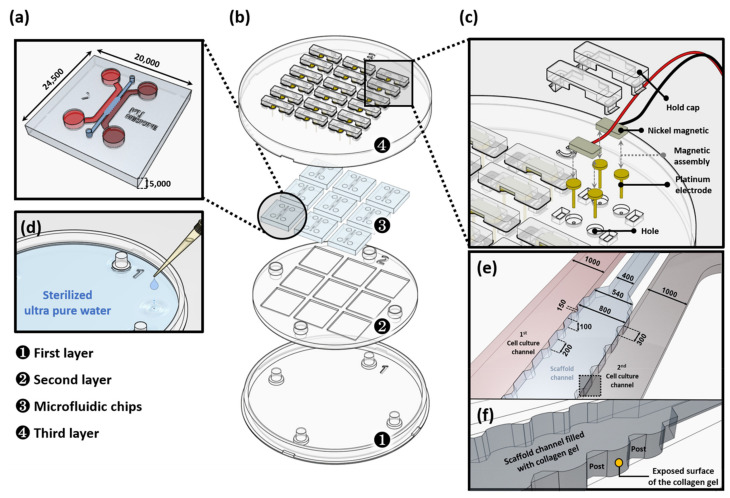
Low constant-current stimulation (LCCS) platform. (**a**) Microfluidic chips with two distinct chambers that are connected through a thin collagen hydrogel channel such that various reagents can diffuse, and a 3D biological structure is formed (e.g., vascular structure) by an interaction between cells and reagents in different chambers. (**b**) Expanded view of the components of the LCCS platform, including co-culture microfluidic chip and the LCCS stimulator. (**c**) Expanded view of the LCCS stimulator. (**d**) Reservoir in the LCCS platform for controlling humidity. (**e**) Schematic of the microfluidic channels including cell culture channels and a scaffold channel. (**f**) Magnified image of the scaffold channel indicated by dotted line box in the (**e**) panel. All scale bars are represented as μm.

**Figure 2 micromachines-12-01291-f002:**
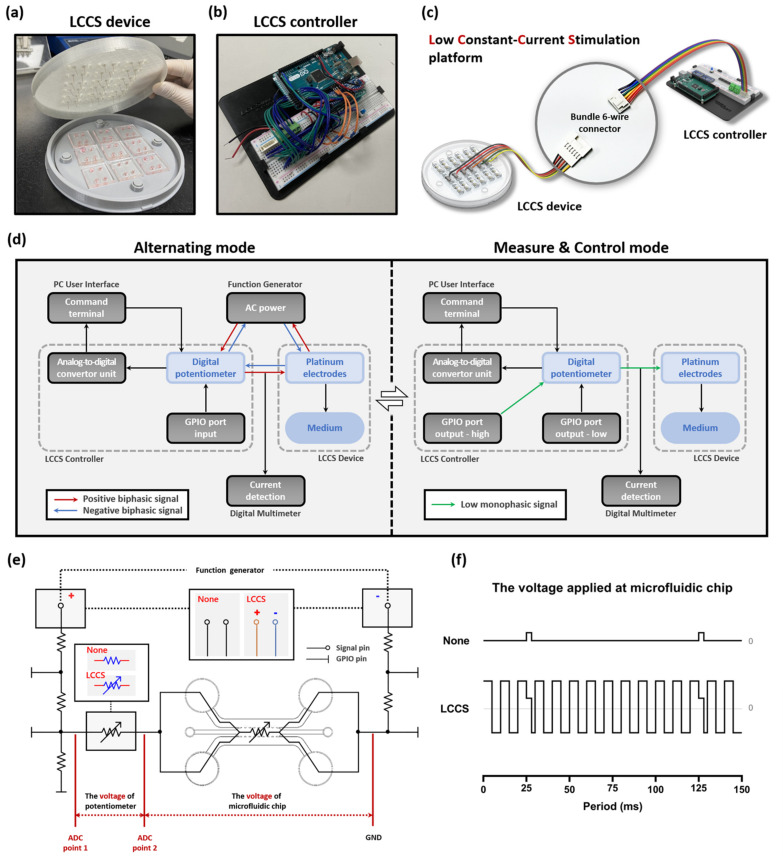
Components of the LCCS platform with inter-operational processes, including the alternating, measure, and control mode. The LCCS platform is composed of (**a**) the LCCS device and (**b**) the controller. (**c**) The LCCS device and controller are connected with bundle 6-wire connectors to send and receive signals. (**d**) The block diagram elucidates the operational process of the LCCS platform and the two modes allowing the alternating, measure, and control modes. (**e**) To analyze the impedance changes and validate the functionality of the LCCS, the transformation of circuits in the controller was conducted under two conditions (None/LCCS). (**f**) Comparison of the voltage waveform for two conditions (None/LCCS). None, condition without constant-current stimulation. LCCS, condition with constant-current stimulation.

**Figure 3 micromachines-12-01291-f003:**
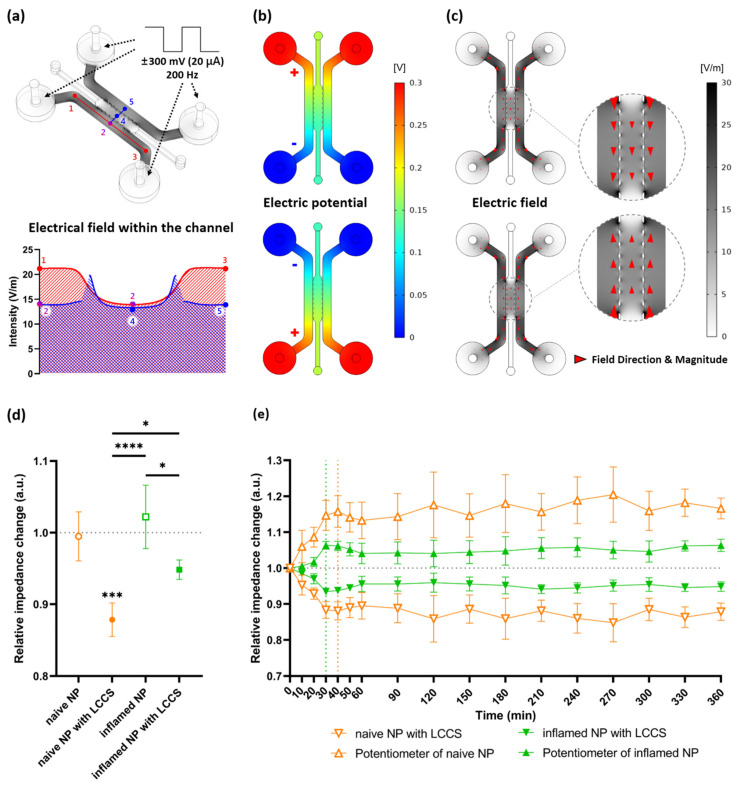
Simulation by applying a biphasic signal to the internal channels and the experimental results of relative impedance change on human NP cells exposed to IL-1β applied at LCCS. (**a**) The electric field within the channels was applied at the biphasic signals (±300 mV, 20 μA, 200 Hz). Each numbered point denotes the electric field intensity. (**b**) Formation of electric potential differences due to polarity (positive and negative) changes of electrodes. (**c**) Changes in electric field direction (red arrowhead) by the different electric potentials. (**d**) Relative impedance changes in the medium from naïve NP or inflamed NP with/without LCCS, and 1.0 denotes the initial impedance (at 0 min), and the relative impedance change was measured after 6 h (360 min) from the initial impedance. (**e**) Relative impedance changes for each group in a time-dependent manner. Each value is the mean ± standard error of four or five independent experiments. *** *p* < 0.001, as compared with naïve NP. * *p* < 0.05, **** *p* < 0.0001, the line indicates comparison with each group.

**Figure 4 micromachines-12-01291-f004:**
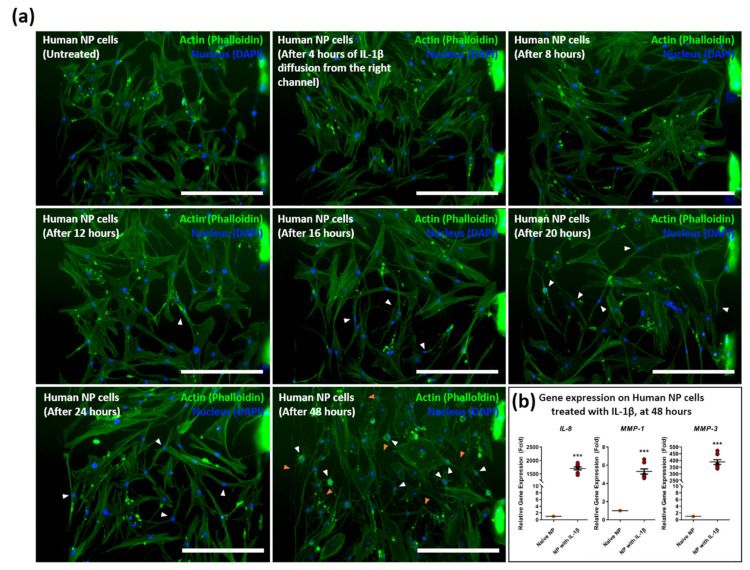
Diffusive stimulation of IL-1β induces inflammatory morphological changes on human NP cells in the microfluidic chip. (**a**) Morphological changes on human NP cells were induced by the IL-1β diffusion gradient from the right channel in a time-dependent manner. Human NP cells exposed to IL-1β show filopodia and lamellipodia extension (orange arrowhead) and cytoplasm shrink (white arrowhead). (**b**) Gene expression of inflammatory mediators, including *IL-8*, *MMP-1*, and *MMP-3* on human NP cells after 48 h of diffusion of IL-1β. Each value is the mean ± standard error of five independent experiments. *** *p* < 0.0001 as compared with naïve NP. Scale bar = 400 μm. Naïve NP; human NP cells cultured in absence of recombinant IL-1β.

**Figure 5 micromachines-12-01291-f005:**
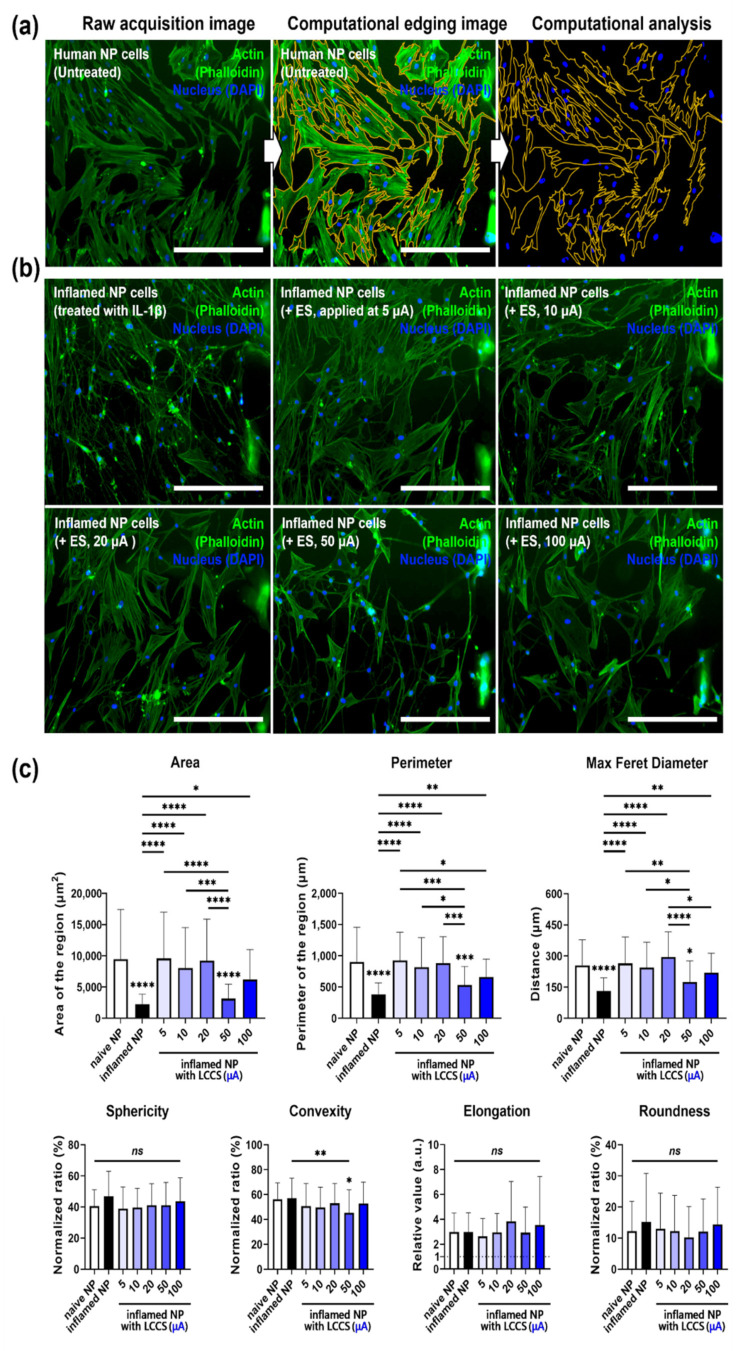
LCCS has a regenerative effect on IL-1β-mediated inflamed human NP cells in the LCCS platform. (**a**) For kinetic analysis, computational edging processing methods were performed from raw acquisition images using imaging software. (**b**) Immunofluorescence image and (**c**) quantitative kinetic measurement of morphological changes in human NP cells, exposed to IL-1 β with/without ES in the LCCS platform. Each value is the mean ± standard error of five independent experiments. Scale bar = 400 μm. *Ns*, no significant difference. * *p* < 0.05, ** *p* < 0.01, *** *p* < 0.001, **** *p* < 0.0001 as compared with naïve NP. The line indicates a comparison with each group.

**Figure 6 micromachines-12-01291-f006:**
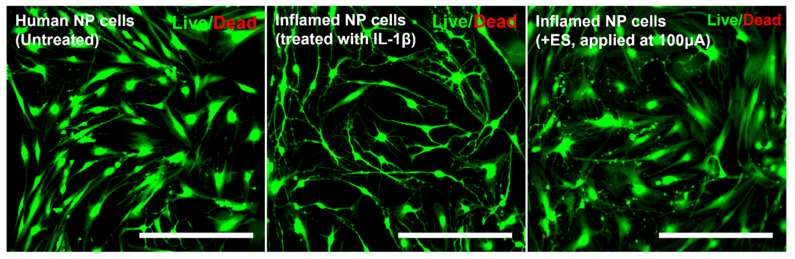
Live and dead viability/cytotoxicity assay depicting the cytotoxic effects of IL-1β or electrical stimulation in human NP cells. Fluorescence images show viable cells stained with acetomethoxy derivate of calcein (green) and nonviable cells stained with ethidium homodimer-1 (red). Scale bar = 400 μm.

## Data Availability

Not applicable.

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
