# Peer review of "Microfluidic Chip with Low Constant-Current Stimulation (LCCS) Platform: Human Nucleus Pulposus Degeneration In Vitro Model for Symptomatic Intervertebral Disc"

_micromachines, 2021, doi:10.3390/mi12111291_

Round 1
Reviewer 1 Report
The authors developed a custom electrical stimulation (ES) platform that can apply low-constant-current stimulation (LCCS) signals to microfluidic chips. The paper is interesting, but I have some suggestions to improve by authors.
1- The dimensions of microfluidic chip is missed.
2- It is not clear that why they used collagen gel in the middle channel. It is concluded from lines 99-101 that they did not encapsulate the cells in collagen, they just cultured the cell on the bottom of the media channel.
3- The introduction should be improved to include all relevant papers.
Reviewer 2 Report
I find the manuscript entitled, " Microfluidic chip with low constant-current stimulation (LCCS) platform: human nucleus pulposus degeneration in vitro model for symptomatic intervertebral disc”, highlights the potential implication of electrical pulse as a therapy for Intervertebral disc degeneration. The microfluidic platform for ES stimulation designed by the author are interesting and enabling controlled electric field for cell stimulation in 2D cell culture model. The authors examined the effects of LCCS on IL-1β-mediated inflammatory NP cells and showed that the constant current with low amplitude and defined frequencies can induce the morphological change of IL-1β-mediated inflammatory NP cells. The authors claim that this morphological change of the NP cells due to electrical stimulation is corelated with IVD regeneration. Besides the cell morphological recovery other molecular aspects of IVD regeneration has not accounted in the current studies. Their findings provide a promising avenue to tune the morphological and biochemical properties of cells by applying the electroceutical cues at a specific window. I include comments below along with other suggestions. However, executing these may require additional work on the part of the authors as I mention.
- In Fig. 4, what fractions of the cells shows the observed change in the morphological properties? The author should show some statistical analysis.
- It is not very clear, how accurately the computational analysis that author used here can threshold each cell boundary properly at the cell dense regions, which may affect on the analysis. The author should show the accuracy validation of the image analysis using cell membrane dye.
- Since IL-1β has a very short half-life, how the authors be sure that the observed effect after 48 hrs due to IL-1β treatment? In Figure 4, author should include a untreated control after 48 hour and corresponding image analysis.
- In Fig.5, the authors only showed the change in cell morphological properties upon expouser of LCCS, but not the other related biochemical changes like qPCR assays in Fig. 4b. It will be necessary to show the gene expressional changes.
- To enhance the implacability of LCCS in the IVD regeneration the author should assess other NP cell characteristics properties including collagen II synthesis, secretion of HIF and IL1.
- The author should explain the mechanism of the LCCS mediated changes in the NP cell morphology. And if it will consider as a electroceuticals how the electrical field induce the IVD regeneration related changes in NP cells.
Reviewer 3 Report
The paper entitled “Microfluidic Chip with Low Constant-Current Stimulation 2 (LCCS) Platform: Human Nucleus Pulposus Degeneration in 3 vitro Model for Symptomatic Intervertebral Disc” presents an interesting usage of the now classic microfluidic device that serves to control the environment of cultured cells. The application of the device to the study of the human nucleus pulposus under treatment with alternative current is remarkable.
While worthwhile the paper presents several weaknesses that preclude its publication. There are lapses in the methodology presented: 1) lack of experimental controls, and 2) lack of statistics for some experiments. Finally, the material and method section is insufficient and some sections would need rewriting for clarity.
In the order of the manuscript:
Material and methods:
- What is the collagen concentration, its buffer?
- How is the device made? Material? Protocol?
- “customized production” is not specific enough, readers should be able to replicate the set-up
- The section between L. 139- l. 156 would benefit from rewriting for better clarity
- “Human NP cells were isolated” from patients? Form cell culture? And how?
- What is the algorithm used to perform image analysis (“computational imaging process”), did the authors develop it? Is it integrated in a software? If so, which one?
- “Quantitative real-time …”: as presented the assay can only be semi-quantitative; even more critically, there is an absence of control gene in the assay. This prevents the accuracy of the analysis reported in the manuscript.
- “Cell cytotoxicity”: using a fluorescent dead-live assay would provide a more direct measure of the effect of the electric current on cell survival
- Figure 3: the caption mentions “Simulation”, but panels d and e are experimental results
- 287: “rapid decrease within 40 min, but inflamed NP was stable after a rapid 287 decrease within 30 min.”, the data do not permit to see a difference in the kinetics
- 291-294: here the logic seemed reversed, isn’t the impedance measured by the adjustment of the potentiometer to keep the current constant?
- Figure 4a: how do untreated NP cells look like after 48 hours incubation on the chip
- Figure 4b: what are the control genes for the real-time RT-PCR?
- Figure 5: how many cells per conditions have been analyzed? The fact that the 50 uA condition exhibits such a difference, and only for morphological parameters, may indicate an outlier; without proper statistics: number of cells , number of replicates, it is difficult to conclude.
Round 2
Reviewer 3 Report
The authors appropriately addressed the initial comments, except for point 7; the set-up used is still not quantitative even with a reference gene; this is a semi-quantitative RT-PCR. A quantitative RT-PCR would involve the generation of calibration curves with known concentrations of the target.
So the you should replace "quantitative" by "semi-quantitative"
Author Response
We sincerely thank the reviewer for the valuable suggestion. We have now replaced "quantitative" by "semi-quantitative", as per your suggestion.